

# Analysis of professional soccer players in competitive match play based on submaximum intensity periods

Eduardo Caro[1,2], Miguel Ángel Campos-Vázquez[3,4],
Manuel Lapuente-Sagarra[5,6,7] and Toni Caparrós[1,8]

[1] National Institute of Physical Education of Catalonia (INEFC), Barcelona, Spain
[2] Neftchi Baku, Baku, Azerbaijan
[3] Watford Football Club, Watford, England
[4] Pablo de Olavide University, Seville, Spain
[5] FC Barcelona, Barcelona, Spain
[6] Department of Physical Education and Sport, University of Basque Country, Vitoria-Gasteiz, Spain
[7] Smart Performance & Sport Science, Cambrils, Spain
[8] GRCE Research Group, National Institute of Physical Education of Catalonia (INEFC), Barcelona, Spain

Corresponding author
Eduardo Caro,
educarobalada@gmail.com

## ABSTRACT

The main objective of this study is to analyse sub-maximum intensity periods (SubMIP's) manifested by professional soccer players during official matches (number of events and time spent in each event), according to the player position, match halve and match, and also to group the players according to their SubMip values during the competition.

**Methods:** We collected a total of 247 individual records of 14 players using Global Positioning System (GPS) during 15 official league matches (Azerbaijan Premier League 2019–2020). We calculated both the number of SubMIPs events and the time each player spent in the SubMIPs zone (threshold of 85% MIP). We analysed the possible independence of the variables with the Kruskal–Wallis test and the possible specific relationships between the groups using a post-hoc analysis with Dunn's test. In order to explore the possible distribution of physical demands in homogeneous groups, a cluster analysis was performed.

**Results:** The statistical analysis showed significant differences between the individual variables in the number of events and in the time spent by the player above the threshold in distance covered at speed >19.8 km/h (HSR), distance covered at speed >25.2 km/h (Sprint), acceleration density (AccDens), mean metabolic power (MetPow), metres per minute (Mmin) and high metabolic load distance >25.5 W/kg (HMLD). Differences were also found according to the playing position in MetPow, Mmin and between halves in AccDens, MetPow, Mmin. In the clustering based on the time spent by the player in SubMIPs, three main groups were described: (1) the centroid was located in lower values in each of the variables; (2) there were an accentuation of the AccDens variable; (3) all the variables, except AccDens, were accentuated.

**Conclusions:** The main differences with regard to SubMIPs were related to the player's individual physical performance and not to position. However, the player's position could act as an attractor and show significant differences during matches.

# INTRODUCTION

Knowledge of players' activity during competitive match play is vitally important for subsequent prescription of specific training (*Riboli, Esposito & Coratella, 2021*). For this purpose, the use of global positioning systems (GPS) offers us valid and reliable information (*Harper, Carling & Kiely, 2019*; *Beato et al., 2018*) on distance and other derived variables (*Gabbett, 2016*), enabling us to study and monitor training and match play (*Casamichanan et al., 2013*; *Gómez-Díaz et al., 2013*).

Previous studies (*Suarez-Arrones et al., 2015*) have analysed the physical activity profile of professional soccer players in competitive match play, reporting the information as mean values (for example, metres per minute or metres sprinted per minute). However, owing to the intermittent nature of play in team sports (*Johnston, Gabbett & Jenkins, 2014*; *Johnston et al., 2018*), these mean values may underestimate the demands to which soccer players are subjected in certain phases of play (*Whitehead et al., 2018*), and if they are used as an intensity benchmark in designing training activities this could lead to insufficient preparation of players (*Riboli, Esposito & Coratella, 2021*). For this reason, in recent years there has been a great increase in research on maximum intensity periods (MIPs), which can be identified as the phases of play in which players show the highest level of conditional exertion. MIP analysis therefore offers us useful information on the maximum demands on athletes (*Lord et al., 2020*). The characteristics of these periods have been analysed in match play (*Casamichana et al., 2019*; *Campos-Vázquez & Lapuente-Sagarra, 2018*), and positional differences have also been evaluated (*Delaney et al., 2018*; *Martín-García et al., 2018*; *Oliva-Lozano et al., 2020*). In addition, recent research has compared the exertion of these periods in various training activities with that displayed by soccer players in competitive matches (*Martín García et al., 2020*; *Martin Garcia et al., 2019*). In these studies, a range of criterion variables, time windows (*Delaney et al., 2018*; *Martín-García et al., 2018*; *Martín García et al., 2020*; *Martin Garcia et al., 2019*) and methods of analysis (*Whitehead et al., 2018*) are used to calculate MIPs, always obtaining higher values when the time window used is smaller (*Casamichana et al., 2019*; *Oliva-Lozano et al., 2020*). Moreover, it seems clear that the demands on athletes are underestimated when segmental analyses are used instead of rolling average techniques (*Campos-Vázquez & Lapuente-Sagarra, 2018*; *Cunningham et al., 2018*) and that these MIPs seem to be context-dependent (*Oliva-Lozano et al., 2020*; *Novak et al., 2021*; *Riboli et al., 2021*).

One of the limitations that arise when calculating MIPs is that they refer to a single event (period) that occurs during play, and therefore do not provide information on time periods during the match when exertion was high, but not maximal. For this reason, it seems useful to undertake an analysis of the external load that soccer players accumulate in these sub-maximum intensity periods (SubMIPs), since they could be related to the levels of fatigue shown in matches (*Schimpchen, Gopaladesikan & Meyer, 2020*). Analyses

**Table 1 Records of training during the research period.**

| | TD (m) | HSR (m) | VHSR (m) | Rel. Dist (m/min) | Acc (count) | Dcc (count) |
|---|---|---|---|---|---|---|
| MD+1 | 4,918.11 ± 694.52 | 157.16 ± 84.01 | 33.01 ± 51.23 | 72.68 ± 10.42 | 51.02 ± 16.07 | 42.87 ± 18.31 |
| MD+1 Rec. | 1,935.23 ± 308.44 | 73.25 ± 68.08 | 8.96 ± 15.32 | 66.79 ± 14.57 | 2.43 ± 2.36 | 2.03 ± 2.31 |
| MD-4 | 5,120.06 ± 835.23 | 216.62 ± 193.25 | 26.71 ± 37.59 | 64.60 ± 13.31 | 46.03 ± 15.77 | 32.11 ± 14.63 |
| MD-3 | 5,716.80 ± 903.00 | 227.38 ± 113.72 | 45.77 ± 41.81 | 69.65 ± 9.07 | 55.85 ± 16.72 | 47.10 ± 16.39 |
| MD-2 | 4,173.90 ± 847.73 | 135.57 ± 124.77 | 21.42 ± 33.89 | 62.59 ± 11.67 | 40.57 ± 12.04 | 30.06 ± 11.76 |
| MD-1 | 2,767.18 ± 563.17 | 69.77 ± 71.64 | 8.10 ± 16.35 | 54.32 ± 7.74 | 31.71 ± 11.73 | 22.99 ± 10.60 |

**Note:**
MD + 1 (Day after the match compensatory work), MD + 1 Rec. (Day after the match recovery work), MD-4 (4 days before the next match), MD-3 (3 days before the next match), MD-2 (2 days before to the next match), MD-1 (1 day before to the next match) TD (Total distance in meters), HSR (Total distance in meters above 19.8 km/h), VHSR (Total distance in meters above 25.2 km/h), Rel. Dist (Relative distance in m/min), Acc (Number of accelerations >3 m/s$^2$), Dcc (Number of deaccelerations <−3 m/s$^2$).

of this kind have previously been performed with internal load variables, such as heart rate (*Aşçı, 2016*), or with other types of methodology that analyse the external load in match play in different ranges of intensity (*Riboli, Esposito & Coratella, 2021*). Moreover, the characteristics of SubMIPs have recently been studied in other team sports such as rugby, Australian rules football (*Johnston et al., 2020*) and futsal, both in competitive match play (*Illa et al., 2020*) and trainings (*Illa et al., 2020*).

The main objective of this study is to analyse SubMIPs manifested by professional soccer players during official matches (number of events and time spent in each event), according to the player position, match halve and match, and also to group the players by cluster grouping according to their SubMip values during the competition.

## METHODS

### Subjects

This study was conducted with 14 professional male soccer players (weight: 73.74 ± 5.92 kg, height: 1.79 ± 0.05 m, age: 23.86 ± 3.58 years), all members of the same team competing in the Azerbaijan Premier League. We analysed 15 official league matches played during the 2019–2020 season. All matches were played in the afternoon, with an interval of at least 5 days between them, with similar microcycles (Table 1).

The playing formation used by the team in official matches was 1-5-3-2, with two midfielders in front of the defensive line and another midfielder behind the forwards. The players were grouped according to their playing position, as central defenders (CD) (*n* = 76 records), wide defenders (WD) (*n* = 50 records), midfielders (MF) (*n* = 36 records), offensive midfielders (OMF) (*n* = 26 records) and forwards (FW) (*n* = 59 records).

In the analysis, the records of players who participated for less than 45 min per half were excluded, as were subjects who did not fulfil the requirement of playing in at least three matches, thereby avoiding atypical values (*Illa et al., 2020*). Those who did not play in the same position throughout these matches were also excluded. We thus obtained a total of 337 individual records, of which 247 met the inclusion criteria.

The people involved in the study gave their written informed consent to use their data for academic purposes. These data were processed following the criteria of the 13th Informed Consent Declaration of Helsinki (*Fortaleza, 2013*) and their use was approved by

the ETHICS COMMITTEE FOR CLINICAL RESEARCH OF THE CATALAN SPORTS COUNCIL witch number 035/CEICGC/2021.

## Instruments

The players used a GPS device (STATSports APEX ProSeries®; STATSports, Newry, Northern Ireland) every day to monitor the external load accumulated in both training sessions and competitive matches. These devices, which operate at a sample rate of 10 Hz configurable to 18 Hz, also include a 600 Hz accelerometer, a 400 Hz gyroscope and a 10 Hz magnetometer: with a weight of 62.7 g and dimensions of $44 \times 84 \times 20$ mm. Recent studies have analysed the validity and inter-unit reliability of these devices, reporting an error of 1–2% for the total distance and maximum speed in team sports (*Beato et al., 2018*). The players wore the device on their upper back, between their shoulder blades, in a vest specially designed for the purpose. The subjects were used to wear the device (*Beato et al., 2018*; *Gimenez et al., 2020*). Furthermore, to ensure appropriate inter-device reliability, the players used the same GPS in all the recordings (*Jennings et al., 2010*), while the data processing and management were performed by the same person, who had a high level of relevant knowledge and experience.

## Procedure

Devices were activated 15 min before the start of the match (*Beato et al., 2018*). In addition, proper connection was checked using the brand's live app (STATSports Apex Live®) during the recordings. Subsequently the raw data from each of the halves of the matches was exported through the manufacturer's software (STATSports® 3.0.03112), using a Microsoft Excel spreadsheet (Microsoft®, Redmond, WA, USA). These recordings were filtered at 10 Hz using a 4th order dual-pass Butterworth filter. The 1-min MIP in each half was then calculated for each player in each of the variables analysed: distance covered at speed >19.8 km/h (high-speed running: HSR), distance covered at speed >25.2 km/h (Sprint), acceleration density (AccDens), mean metabolic power (MetPow), metres per minute (Mmin) and high metabolic load distance >25.5 W/kg (HMLD), as in previous studies (*Martín-García et al., 2018*; *Martin Garcia et al., 2019*; *Di Mascio & Bradley, 2013*; *Delaney et al., 2018*). In addition, we applied a threshold of 85% of the individual mean of the three highest MIPs shown by each participant in order to delimit the range of activity performed in the SubMIP zone (*Illa et al., 2020*). For the simple variables, such as distance or distance at a certain speed, no extraordinary calculation was necessary beyond applying the individual 85% of each of the players each game each period, every time that 85% was exceeded, the counting of the event and the sum of seconds began. Complex variables, such as HMLD or MetPow, were used in the same way that they have been used in previous studies (*Martín-García et al., 2018*; *Martin Garcia et al., 2019*; *Di Mascio & Bradley, 2013*; *Delaney et al., 2018*), to subsequently apply the 85% MIP.

## Statistical analysis

A central trend descriptive analysis was performed, subsequently analysing the normality of the variables studied with the Shapiro–Wilk test. In view of the non-normality of the

sample, we analysed the possible independence of the variables with the Kruskal–Wallis test and the possible specific relationships between the groups using a post-hoc analysis with Dunn's test (*López-Roldán & Fachelli, 2015*). In order to explore the possible distribution of physical demands in homogeneous groups, a cluster analysis was performed. To determine the number of clusters, a hierarchical cluster analysis (HCA) was performed, standardizing the sample values (Z-score) beforehand. The variables were clustered in the groups obtained using the k-mean method. Once the clusters had been established, possible associations were determined with an ANOVA analysis (*Izquierdo et al., 2019*). The Z-score was used as the criterion to establish the value of the dimensions as high, moderate or low. Values between –0.5 and +0.5 standard deviations (SDs) around the standardized mean were considered moderate, scores greater than +0.5 SDs high and scores below −0.5 SDs low (*Izquierdo et al., 2019*). The statistical analysis was performed with SPSS software (Statistics for Windows version 25; IBM Corp., NY, USA). The significance level in all cases was $p < 0.05$.

## RESULTS

The results are shown as mean plus/minus standard deviation (Table 2). The highest average values SubMip were found in the variables AccDens and Mmin, and the lowest values in Sprint.

Those who exceeded the threshold in the HSR variable on the most occasions and for the longest time were MF and WD; by contrast, CD showed the lowest values in number of events and time above the threshold in the HSR, Sprint and HMLD variables. WD had the highest number and duration values in Sprint and HMLD. OMF showed higher values than the other positions in number of events and time spend above the threshold in the AccDens variable, as opposed to FW, who had the lowest values in this variable. In the MetPow and Mmin variables WD were those with the lowest time and duration values. The position that showed the highest values in these two variables was MF. The average values for number of AccDens events and duration of Mmin proved to be the highest in the first half of the match, and the values for number of Sprint events and duration in HSR were the lowest during the second half (Table 2).

The statistical analysis showed significant differences between the individual variables in the number of events in HSR (H = 27.805, $p = 0.01$), AccDens (H = 51.733, $p < 0.001$), MetPow (H = 74.44, $p < 0.001$) and Mmin (H = 66.751, $p < 0.001$) and in the time the player spent above the threshold in the same variables (HSR: H = 26.7, $p = 0.014$; AccDens: H = 49.455, $p < 0.001$; MetPow: H = 68.868; $p < 0.001$; Mmin: H = 63.655, $p < 0.001$) (Table 3). Significant differences were also observed between the variables according to the playing position for number of events in MetPow (H = 50.55, $p < 0.001$) and Mmin (H = 44.099, $p < 0.001$) and the time above the threshold in these variables (MetPow: H = 47.888, $p < 0.001$; Mmin: H = 42.328; $p < 0.001$) (Table 3). Similarly, differences were found between halves in number of events for AccDens (H = 5.797, $p = 0.016$), MetPow (H = 7.402, $p = 0.007$), Mmin (H = 6.05, $p = 0.014$) and time above the threshold in these variables (AccDens: H = 8.611, $p = 0.003$; MetPow: H = 8.068, $p = 0.005$; Mmin: H = 8.602, $p = 0.003$). Only were found differences between matches for events and time

**Table 2 Mean and standard deviation of 247 events recorded during the 15 games, differentiated by position and halves.**

| Position | CD | | | WD | | | MF | | | OMF | | | FW | | | | | |
|---|---|---|---|---|---|---|---|---|---|---|---|---|---|---|---|---|---|---|
| Half | 1st (n=39) | 2nd (n=37) | Total (n=76) | 1st (n=29) | 2nd (n=21) | Total (n=50) | 1st (n=19) | 2nd (n=17) | Total (n=36) | 1st (n=15) | 2nd (n=11) | Total (n=26) | 1st (n=36) | 2nd (n=23) | Total (n=59) | 1st (n=138) | 2nd (n=109) | 1st and 2nd (n=247) |
| | Mean ± SD | Mean ± SD | Mean ± SD | Mean ± SD | Mean ± SD | Mean ± SD | Mean ± SD | Mean ± SD | Mean ± SD | Mean ± SD | Mean ± SD | Mean ± SD | Mean ± SD | Mean ± SD | Mean ± SD | Mean ± SD | Mean ± SD | Mean ± SD |
| # HSR | 0.103 ± 0.307 | 0.162 ± 0.374 | 0.132 ± 0.34 | 0.276 ± 0.455 | 0.190 ± 0.402 | 0.240 ± 0.431 | 0.263 ± 0.562 | 0.235 ± 0.562 | 0.250 ± 0.554 | 0.313 ± 0.479 | 0.100 ± 0.316 | 0.231 ± 0.430 | 0.278 ± 0.566 | 0.130 ± 0.458 | 0.220 ± 0.527 | 0.23 ± 0.471 | 0.167 ± 0.421 | 0.202 ± 0.45 |
| Duration HSR | 0.151 ± 0.460 | 0.284 ± 0.665 | 0.216 ± 0.57 | 0.429 ± 0.725 | 0.305 ± 0.660 | 0.377 ± 0.694 | 0.429 ± 0.903 | 0.277 ± 0.685 | 0.357 ± 0.800 | 0.445 ± 0.701 | 0.176 ± 0.556 | 0.341 ± 0.651 | 0.384 ± 0.766 | 0.233 ± 0.809 | 0.325 ± 0.78 | 0.341 ± 0.699 | 0.266 ± 0.681 | 0.308 ± 0.691 |
| # Sprint | 0.077 ± 0.270 | 0.135 ± 0.347 | 0.105 ± 0.309 | 0.172 ± 0.384 | 0.238 ± 0.539 | 0.2 ± 0.452 | 0.211 ± 0.419 | 0.117 ± 0.332 | 0.167 ± 0.378 | 0.188 ± 0.544 | 0.100 ± 0.316 | 0.154 ± 0.464 | 0.222 ± 0.422 | 0.130 ± 0.344 | 0.186 ± 0.393 | 0.165 ± 0.392 | 0.148 ± 0.382 | 0.158 ± 0.387 |
| Duration Sprint | 0.150 ± 0.528 | 0.262 ± 0.673 | 0.205 ± 0.602 | 0.328 ± 0.731 | 0.461 ± 1.046 | 0.384 ± 0.87 | 0.386 ± 0.775 | 0.230 ± 0.649 | 0.313 ± 0.713 | 0.364 ± 1.056 | 0.197 ± 0.662 | 0.299 ± 0.903 | 0.363 ± 0.713 | 0.250 ± 0.661 | 0.319 ± 0.69 | 0.299 ± 0.723 | 0.287 ± 0.742 | 0.294 ± 0.73 |
| # AccDens | 2.692 ± 2.028 | 1.946 ± 1.393 | 2.329 ± 1.777 | 2.207 ± 1.521 | 1.619 ± 1.284 | 1.96 ± 1.442 | 2.421 ± 2.168 | 2.411 ± 2.123 | 2.417 ± 2.116 | 2.938 ± 1.948 | 2.000 ± 2.108 | 2.577 ± 2.023 | 1.694 ± 0.822 | 1.696 ± 1.146 | 1.695 ± 0.951 | 2.324 ± 1.729 | 1.907 ± 1.532 | 2.142 ± 1.655 |
| Duration AccDens | 3.385 ± 3.352 | 2.011 ± 2.279 | 2.716 ± 2.943 | 2.617 ± 2.278 | 1.751 ± 2.080 | 2.254 ± 2.217 | 3.359 ± 3.731 | 2.750 ± 3.155 | 3.072 ± 3.436 | 4.011 ± 3.135 | 2.569 ± 2.967 | 3.457 ± 3.095 | 2.013 ± 1.286 | 1.749 ± 1.858 | 1.91 ± 1.525 | 2.938 ± 2.813 | 2.073 ± 2.374 | 2.56 ± 2.66 |
| # MetPow | 1.513 ± 1.144 | 1.108 ± 1.430 | 1.316 ± 1.298 | 0.759 ± 1.091 | 0.905 ± 1.546 | 0.82 ± 1.289 | 3.579 ± 1.742 | 2.647 ± 1.538 | 3.139 ± 1.693 | 1.688 ± 1.302 | 0.900 ± 1.287 | 1.385 ± 1.329 | 2.194 ± 1.582 | 1.348 ± 1.191 | 1.864 ± 1.491 | 1.835 ± 1.595 | 1.343 ± 1.505 | 1.619 ± 1.572 |
| Duration MetPow | 2.007 ± 1.560 | 1.400 ± 1.789 | 1.712 ± 1.698 | 0.975 ± 1.391 | 1.125 ± 1.895 | 1.039 ± 1.606 | 4.736 ± 2.398 | 3.370 ± 1.913 | 4.091 ± 2.261 | 2.231 ± 1.551 | 1.214 ± 1.756 | 1.840 ± 1.676 | 2.734 ± 1.971 | 1.617 ± 1.395 | 2.299 ± 1.84 | 2.379 ± 2.077 | 1.686 ± 1.876 | 2.076 ± 2.017 |
| # m/min | 1.615 ± 1.330 | 1.459 ± 1.966 | 1.539 ± 1.661 | 1.000 ± 1.363 | 0.952 ± 1.284 | 0.98 ± 1.317 | 3.632 ± 1.921 | 2.823 ± 1.740 | 3.250 ± 1.857 | 2.875 ± 2.187 | 1.200 ± 1.619 | 2.231 ± 2.122 | 2.667 ± 1.882 | 1.870 ± 1.359 | 2.356 ± 1.73 | 2.18 ± 1.885 | 1.639 ± 1.737 | 1.943 ± 1.838 |
| Duration m/min | 2.242 ± 1.844 | 1.825 ± 2.536 | 2.04 ± 2.203 | 1.433 ± 2.293 | 1.187 ± 1.630 | 1.33 ± 2.026 | 4.782 ± 2.569 | 3.562 ± 2.405 | 4.206 ± 2.534 | 3.958 ± 2.680 | 1.665 ± 2.353 | 3.076 ± 2.756 | 3.497 ± 2.441 | 2.408 ± 1.750 | 3.073 ± 2.246 | 2.944 ± 2.537 | 2.084 ± 2.278 | 2.568 ± 2.46 |
| # HMLD (count) | 0.231 ± 0.536 | 0.243 ± 0.495 | 0.237 ± 0.513 | 0.345 ± 0.614 | 0.333 ± 0.483 | 0.34 ± 0.557 | 0.316 ± 0.582 | 0.294 ± 0.587 | 0.306 ± 0.577 | 0.438 ± 0.629 | 0.100 ± 0.316 | 0.308 ± 0.549 | 0.333 ± 0.478 | 0.130 ± 0.458 | 0.254 ± 0.477 | 0.317 ± 0.552 | 0.231 ± 0.485 | 0.279 ± 0.525 |
| Duration HMLD | 0.300 ± 0.707 | 0.347 ± 0.705 | 0.323 ± 0.702 | 0.446 ± 0.800 | 0.521 ± 0.768 | 0.477 ± 0.78 | 0.422 ± 0.762 | 0.349 ± 0.759 | 0.388 ± 0.751 | 0.564 ± 0.826 | 0.173 ± 0.547 | 0.414 ± 0.745 | 0.480 ± 0.712 | 0.183 ± 0.633 | 0.365 ± 0.693 | 0.424 ± 0.744 | 0.33 ± 0.697 | 0.383 ± 0.724 |

**Notes:**

Number of actions (#) and duration of these (minutes) when the threshold of 85% of the maximum values previously stipulated is exceeded. The ranges and units of measurement for each variable were as follows. HSR meters at speed >19.8 km/h, Sprint meters at speed >25.2 km/h, AccDens is the average accelerations of the absolute values (m/s$^2$) during the duration of the event, MetPow values in the form of W·kg, m/min meters run per minute and HMLD running meters at high metabolic power (>25.5 W·kg).

Centre-defenders (CD), Wide-defenders (WD), Midfielders (MF), Offensive-Midfielders (OMF), Forwards (FW).

**Table 3 Kruskal–Wallis test.**

| Variable | Factor | Individual | Position | Halves | Match |
|---|---|---|---|---|---|
| # HSR | St. H | 27.805* | 2.685 | 1.448 | 12.637 |
| Duration HSR | St. H | 26.7* | 2.34 | 1.199 | 13.413 |
| # Sprint | St. H | 13.669 | 2.43 | 0.17 | 11.146 |
| Duration Sprint | St. H | 12.509 | 1.811 | 0.072 | 10.872 |
| #AccDens | St. H | 51.733** | 3.937 | 5.797* | 14.063 |
| Duration AccDens | St. H | 49.455** | 4.537 | 8.611* | 16.528 |
| # MetPow | St. H | 74.44** | 50.55** | 7.402* | 22.723 |
| Duration MetPow | St. H | 68.868** | 47.888** | 8.068* | 26.513* |
| #m/min | St. H | 66.751** | 44.099** | 6.05* | 30.971* |
| Duration m/min | St. H | 63.655** | 42.328** | 8.602* | 32.582* |
| #HMLD | St. H | 15.968 | 1.747 | 1.637 | 22.072 |
| Duration HMLD | St. H | 14.293 | 1.646 | 1.376 | 21.147 |

**Notes:**
* $p < 0.05$.
** $p < 0.001$.
Number of actions (#) and duration of these (minutes) when the threshold of 85% of the maximum values previously stipulated is exceeded.

of Mmin (H = 30.971, $p$ = 0.006, H = 32.582, $p$ = 0.003) and in time of MetPow (H = 26.513, $p$ = 0.022) (Table 3).

The subsequent post-hoc analysis showed differences between OMF and FW in AccDens time, also shows significant differences in the MetPow and m/min variables in number of events and duration between all positions except between OMF and FW, between OMF and CD differences were found only in the duration m/min (Table 4).

Analysing the first and second halves by post-hoc analysis, we can see differences both in the number of events of AccDens, MetPow and Mmin (Z = 2.408, $p$ = 0.008; Z = 2.721, $p$ = 0.003; Z = 2.46, $p$ = 0.007 respectively) and in time above the threshold in these same variables (Z = 2.934, $p$ = 0.002; Z = 2.84, $p$ = 0.002; Z = 2.933, $p$ = 0.002 respectively), values for the first half being higher than for the second (Table 4).

In the clustering based on the time spent by the player in SubMIPs, three main groups were described (Fig. 1). In the first, the centroid was located in highest values in HSR, Sprint and HMLD. In the second, an accentuation of the AccDens, MetPow and Mmin variables was observed, and in the third and final cluster the centroid was located in lower values in each of the variables (Table 5).

When these clusters are compared to the positions used for the analysis, we can see behavioural tendencies during match play. In all positions except midfielders, we found a tendency to the third cluster (CD (68.4%), WD (74%), OMD (57.7%) and FW (64%)), MF tend to group 2 (50%). For CD, OMF and FW, the second option is group 2 (19.6%, 23.1% and 18.6% respectively), while for WD, group 1 (22%).

The ANOVAs revealed significant differences between the three clusters ($p < 0.01$), and the Levene F value indicated significant differences between the three clusters for each variable analysed ($p < 0.01$). The sampling adequacy measure was KMO = 0.64, for $X^2$ = 537.0 and $p$ = 0.00 (*Morissette & Chartier, 2013*).

**Table 4 Dunn's *Post Hoc* comparisons between positions and halves.**

| Variable | St. | MF-OMF | MF-CD | MF-WD | MF-FW | OMF-CD | OMF-WD | OMF-FW | CD-WD | CD-FW | WD-FW | 1°-2° Half |
|---|---|---|---|---|---|---|---|---|---|---|---|---|
| | | **Positions** | | | | | | | | | | **Halves** |
| #HSR | St. Z | −0.254 | 0.890 | −0.406 | 0.308 | 1.081 | −0.096 | 0.555 | −1.474 | −0.662 | 0.801 | 1.203 |
| Duration HSR | St. Z | −0.225 | 0.804 | −0.419 | 0.280 | 0.971 | −0.139 | 0.498 | −1.397 | −0.596 | 0.785 | 1.095 |
| #Sprint | St. Z | 0.490 | 0.841 | −0.207 | −0.259 | 0.194 | −0.709 | −0.768 | −1.184 | −1.297 | −0.049 | 0.412 |
| Duration Sprint | St. Z | 0.530 | 0.854 | −0.068 | −0.044 | 0.160 | −0.626 | −0.619 | −1.030 | −1.050 | 0.028 | 0.268 |
| #AccDens | St. Z | −0.636 | −0.258 | 0.552 | 1.022 | 0.490 | 1.176 | 1.614 | 0.949 | 1.547 | 0.497 | 2.408* |
| Duration AccDens | St. Z | −0.888 | 0.256 | 0.714 | 1.128 | 1.233 | 1.590 | 1.984* | 0.573 | 1.077 | 0.430 | 2.934* |
| #MetPow | St. Z | 3.848** | 5.129** | 6.752** | 3.148** | 0.208 | 2.008* | −1.379 | 2.406* | −2.144* | −4.215** | 2.721* |
| Duration MetPow | St. Z | 3.644** | 5.034** | 6.663** | 3.321** | 0.354 | 2.144* | −1.001 | 2.405* | −1.822* | −3.923** | 2.840* |
| #m/min | St. Z | 2.318* | 4.655** | 5.897** | 2.011* | 1.519 | 2.863* | −0.728 | 1.906* | −2.977* | −4.493** | 2.460* |
| Duration m/min | St. Z | 1.885* | 4.408** | 5.732** | 1.855* | 1.790* | 3.175** | −0.394 | 1.983* | −2.879* | −4.477** | 2.933* |
| #HMLD | St. Z | −0.129 | 0.625 | −0.470 | 0.237 | 0.703 | −0.288 | 0.354 | −1.259 | −0.440 | 0.795 | 1.280 |
| Duration HMLD | St. Z | −0.229 | 0.444 | −0.629 | 0.030 | 0.655 | −0.325 | 0.277 | −1.249 | −0.481 | 0.748 | 1.173 |

Notes:
* $p < 0.05$.
** $p < 0.001$.
Number of actions (#) and duration of these (minutes) when the threshold of 85% of the maximum values previously stipulated is exceeded. Centre-defenders (CD), Wide-defenders (WD), Midfielders (MF), Offensive-Midfielders (OMF), Forwards (FW).

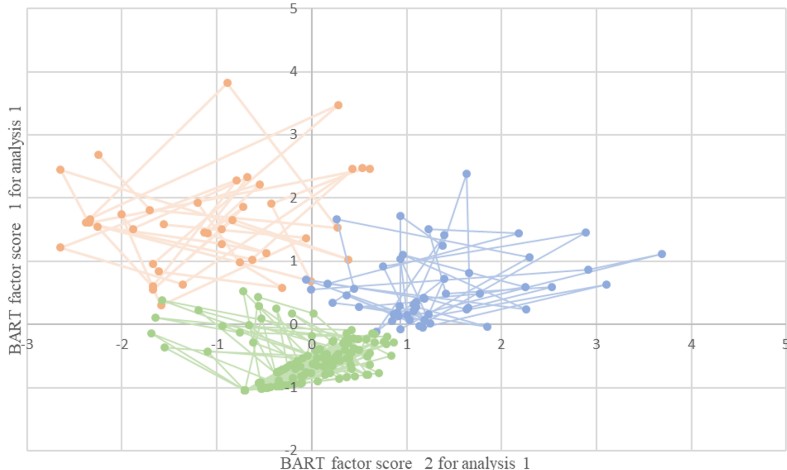

**Figure 1 Groups based on the time the players are in % > the 85% threshold.**

## DISCUSSION

The most important findings of this study are as follows: (I) the largest differences in physical demands based of SubMIPs in competitive matches are individual (HSR, AccDens, MetPow and Mmin); (II) there are differences between positions, between halves and between matches in the number of SubMIP events and the duration of these events in different variables; (III) the largest differences between positions are found in the MetPow and Mmin variables, while the differences between halves are most evident for

**Table 5 Clustering centroid in each of the variables for each group.**

|  | 1 | 2 | 3 |
|---|---|---|---|
| Z-Score duration HSR | 2.00864 | −0.41848 | −0.39346 |
| Z-Score duration Sprint | 0.96536 | −0.06218 | −0.23602 |
| Z-Score duration AccDens | −0.04516 | 0.64905 | −0.20714 |
| Z-Score duration MetPow | 0.65727 | 1.16692 | −0.56901 |
| Z-Score duration m/min | 0.56242 | 1.19646 | −0.55373 |
| Z-Score duration HMLD | 1.71555 | −0.10788 | −0.42031 |

AccDens, MetPow and Mmin; (IV) the physical demand is not determined solely by the position of a player, although this determines his way of acting during matches.

Recent research (*Riboli, Esposito & Coratella, 2021*) analysed the percentage distribution of intensity compared to MIP 1', concluding that the average data in this distribution were much lower that the MIP 1' values, especially for high-intensity exertion (9–19%). This type of analysis has also been considered in sports such as rugby and Australian rules football (*Johnston et al., 2020*), as well as in much more closely related sports such as futsal, both in match play and in training (*Illa et al., 2020*; *Illa et al., 2020*). The study cited (*Illa et al., 2020*) showed that high-intensity (80%) and very high-intensity (90%) exertions in competition were lower in high-speed actions (in this case only HSR) (0.17 ± 0 in both categories) and higher in distance variables (3.17 ± 2.32 and 0.75 ± 0.35 respectively) and acceleration variables (2 ± 1.12 and 0.67 ± 0.55 respectively). As these are different sports, these results cannot be compared to those of our study.

To the best of our knowledge, this is the first study to date that has analysed the number of events and the time spent in SubMIP in professional soccer matches. The most notable results at the individual level show significant differences ($p < 0.01$) between players in HSR, AccDens, MetPow and Mmin events, as well as in the duration of each. These results suggest that we should consider the physical demands of players in competitive match play individually. Also extrapolating this to training sessions, individualizing them, and considering SubMIP as one more tool for the control of the load.

The differences in physical demands between positions in match play have been extensively studied with classical variables analysis using absolute speed thresholds (*Mallo et al., 2015*; *Di Salvo et al., 2007*) and individual relative thresholds (*Abbott, Brickley & Smeeton, 2018*) or with analysis of MIPs in various time windows (*Martín-García et al., 2018*; *Di Mascio & Bradley, 2013*). Regardless of the method used, all these studies showed significant differences between positions. In our study, analysing of SubMIPs, we found significant differences in the AccDens variable (only in its duration), which was higher in OMF than in FW. The greatest differences between positions occur in the MetPow and Mmin variables in number of events (Table 3), except between OMF in comparison with FW and CD. These differences could be partially explained by position-dependent nature in soccer (*Martín-García et al., 2018*; *Di Mascio & Bradley, 2013*; *Mallo et al., 2015*; *Di Salvo et al., 2007*; *Abbott, Brickley & Smeeton, 2018*); indeed, using individualized threshold

values for the SubMIP based on the MIP for each player, we can find significant differences between positions in the number of events and the time in SubMIP.

The results obtained show higher values in the first half than in the second for AccDens, MetPow and Mmin, in number of events (Table 3), with no significant differences between halves in the other variables analysed. Previous studies also showed differences between halves when performing an analysis using classical variables, a lowering of total distance (*Di Salvo et al., 2007*) and a reduction of the distance covered in various speed ranges in the second half (*Rampinini et al., 2007*), though not systematically. Moreover, when MIPs were analysed, differences were found in variables such as average metabolic power, and these differences were more evident when the time window analysed was larger (*Casamichana et al., 2019*). Despite the apparent reduction in physical performance in the second half shown by these studies, we should treat this information with caution, as the reduction in performance could be a result of the players' accumulated fatigue, but it could also be due to the reduction in useful playing time (*Rey et al., 2011*) or other reasons.

On the other hand, there were almost no differences in of SubMIPs when a comparison was made between matches. Significant differences were only apparent in the duration of MetPow and in number of events and duration of Mmin. This is contrary to what is reported in previous studies that have analysed the same phenomenon from a traditional perspective, where greater differences are shown, especially in metres covered at high intensity (*Oliva-Lozano et al., 2020*). The individualization of the thresholds for analysis of the number of events and duration could explain this absence of differences between matches.

Clustering produced three major behavioural groups. The players of different positions are grouped in group 3, where the demands in all the variables are lower, suggesting that players who, for various reasons, have not undergone great conditional exertion during a match are located here, considering the number of SubMIP events during the competition, it seems normal. In the first group, the centroid was located in highest values in HSR, Sprint and HMLD, we could call it the sprinter group, WDs are placed as a second option.

Finally, group number 2, an accentuation of the AccDens, MetPow and Mmin variables was observed, and the MF are placed in it as the first option and the CD, OMF and FW as the second option.

An important factor to take into account when interpreting the results is the methodology used to calculate SubMIPs. In our study, the average of the three highest MIPs for each of the variables was individual, and therefore so were the thresholds for the SubMIP. This means that a player who obtains higher maximum peaks in a specific variable will have higher SubMIP thresholds than another player with lower MIP values, giving different possibilities in the relationship between MIP and SubMIP, depending on the variable and the player's physical demands during the match. On this context, there are players with high MIP and many SubMIP events, for example, WD in HSR, or both low values like CD in Sprint as the nature of their position requires them to use or not certain demands. Players with lower MIP and higher SubMIP periods (such as CD in AccDens, probably because of the low SubMIP threshold) and players with high MIP and low

SubMIP, are rarely able to exceed the SubMIP range, either due to the low frequency with which the actions are requested or due to the very high threshold at which these actions are found.

It is important to bear in mind that an athlete's conditional demands in competitive play, and consequently the MIP attained, is multifactorial (*Novak et al., 2021*); this is why the average of the three highest MIPs for each player in each variable was taken, so as to minimize the possible noise that may be caused in the MIP by context (*Illa et al., 2020*). Given that players exceed the SubMIP theshold approximately 0.2 times per half in HSR, Sprint and HMLD, we can see that MIP analysis from a non-dynamic perspective, analysing matches independently, may underestimate the real maximum capacities of our athletes. An analysis over the course of different matches and training sessions may offer us a view closer to the real maximum demand of players, therefore, it could be useful to standardiHe the criteria for obtaining SubMIPs and explore this field in greater depth so as to obtain more detailed knowledge of player's physical demands from this new perspective, with the aim of providing valid and reliable tools for designing training and analysing their conditional performance in competitive match play.

However, main limitations of this study were the sample size (a single professional team) and the number of matches analysed, so these results should be interpreted according to this specific competitive context.

## CONCLUSIONS

In conclusion, the analysis of the SubMIP in professional soccer players during official matches shows individual-dependent differences in the variables HSR, AccDens, MetPow and Mmin. There are also differences between positions in MetPow and Mmin; AccDens, MetPow and Mmin for halves; and MetPow and Mmin for matches. Therefore, we can conclude that physical demands are not determined solely by a player's position although their position may influence their playing profile during matches.

## PRACTICAL APPLICATIONS

Just as the use of MIP for monitoring load and analysing match play has brought about a change in training design, the use of SubMIPs to discover and quantify the physical demands of professional soccer players may be a useful tool that could help to optimiHe players' performance.

Bearing in mind the individual differences in the characteristics of SubMIPs during match play in professional soccer, it seems appropriate to individualized training according to the profile of each player and to the relationship of MIP to SubMIP profiles, as this could be an indicator of the physical demand required and whether this leads to the desired adaptations in the athletes. In the same way, we can find different profiles of athletes, as we have been able to see in the groupings and in the differences of the possible MIP-SubMIP relationships. It seems interesting to also take this into account for the design of training sessions.

### Funding

The authors received no funding for this work.

### Competing Interests

The authors declare that they have no competing interests. Eduardo Caro is employed by and Nefcthi futbol klubu, Baku and Manuel Lapuente-Sagarra is employed by FC Barcelona and Smart Performance & Sport Science

### Author Contributions

- Eduardo Caro conceived and designed the experiments, performed the experiments, analyzed the data, prepared figures and/or tables, and approved the final draft.
- Miguel Ángel Campos-Vázquez conceived and designed the experiments, performed the experiments, authored or reviewed drafts of the paper, and approved the final draft.
- Manuel Lapuente-Sagarra conceived and designed the experiments, performed the experiments, authored or reviewed drafts of the paper, and approved the final draft.
- Toni Caparrós performed the experiments, analyzed the data, authored or reviewed drafts of the paper, and approved the final draft.

### Human Ethics

The following information was supplied relating to ethical approvals (*i.e.*, approving body and any reference numbers):

The ethics committee of the Clinical Research of the Catalan Sports Council (035/CEICGC/2021).

### Data Availability

The raw data is available at figshare: Caro, Eduardo (2022): RAW DATA GAMES .xlsx. figshare. Dataset. https://doi.org/10.6084/m9.figshare.17105060.v1.

### Supplemental Information

Supplemental information for this article can be found online at http://dx.doi.org/10.7717/peerj.13309#supplemental-information.

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
