# Peer review of "Analysis of professional soccer players in competitive match play based on submaximum intensity periods"

_PeerJ, doi:10.7717/peerj.13309_

## Round 0.1 · original submission · Minor Revisions

The two reviewers and I are impressed with the study and the manuscript you submitted. However, the two reviewers have still identified a number of additional areas in which the manuscript can be improved. Please take on board the feedback of the reviewers in your resubmission.

Reviewer 1 ·

Basic reporting

English is not my mother tongue. I cannot judge the correctness of the English language used. On the other hand, the scientific language used by the authors is correct in my opinion.

The introduction to the subject is, in my opinion, on a good level. As in any article, you can indicate corrections resulting from your own subjective view of the problem under study. In this aspect, I want to highlight three aspects:
1) there is an extensive statistical method in the article and the purpose of the article is very simple. You can consider introducing brief information about the applied statistical method (hierarchical cluster analysis) to the target. It will inform you from the very beginning that these are not simple comparative analyzes.
2) due to the variety and number of the described parameters and factors, in the introduction, I lack information about the tests performed in the context of all variables used in the work (e.g. high metabolic load distance, etc.).
3) In my opinion, the discussion is conducted with little use of a variety of references. Duplicate results are a large part of the discussion. The results are discussed with little problem depth. (e.g. there is no discussion of mean metabolic power and others as an important parameter). In addition, Line 315-336 - this is a partial description of the method used, consider moving some of this information to the Methods section.

The structure of the article is correct. However, I believe that the description of the results could be shorter and some of the calculations, e.g. post-hoc, could be shown in the table.
Tables do not include units next to parameters (eg table 2, eg Duration Sprint in what units? Etc.). All abbreviations used in the tables should be clearly explained (e.g. table 3 - factor - st.?).

Experimental design

In my opinion, the methods should be supplemented with the definitions of the parameters used in the article. I understand that they are collected using specific hardware, but the reader needs to know what it means and how it is calculated, e.g. metabolic load distance and other indicators.

Moreover, in the current research describing positions on the pitch, it is no longer possible to show only 4 positions on the pitch. There is a big difference in the actions of different midfielders and they should be distinguished as different positions (side, defensive, offensive, etc.).

Validity of the findings

The article describes a topic often found in the literature, especially through the increasingly available modern external load testing tools. However, what's new is that the results are shown in the context of the SubMIPs zone (threshold of 85% MIP). Unfortunately, there is no detailed calculation of the threshold values and individual thresholds.

·

Basic reporting

Clear and unambiguous, professional English used throughout.
Yes

Literature references, sufficient field background/context provided.
Yes

Professional article structure, figures, tables. Raw data shared.
Yes

Self-contained with relevant results to hypotheses.
Yes

Experimental design

Original primary research within Aims and Scope of the journal.
Manuscript in the field of performance analysis, sports science

Research question well defined, relevant & meaningful. It is stated how research fills an identified knowledge gap.

Yes
Rigorous investigation performed to a high technical & ethical standard.
Yes

Methods described with sufficient detail & information to replicate.
Rather yes

Validity of the findings

Impact and novelty not assessed. Meaningful replication encouraged where rationale & benefit to literature is clearly stated.
The research brings interesting information to the literature

All underlying data have been provided; they are robust, statistically sound, & controlled.
Rather yes

Conclusions are well stated, linked to original research question & limited to supporting results.
Yes

Additional comments

The topic is not very revealing, but it is in line with the current trends in the analysis of performance in football
In the abstract, in the method section, line 37, there is no sentence informing you which statistical analysis you used to present the results. Please add
Introduction is well-written and justified
Methods. Line 91, Azerbaijan Premier League is the highest level of competition in this country?
Line 99, One of the biggest my concerns regarding division of playing positions. Midfielders have different task when playing in central or in wide side of the pitch. this can distort the data analysis
In this type of analysis, the most common division is into five playing positions, not four
Line 102-106 fine
Results. Line 159-161 First sentence of the results is misleading, I can not understand second part. Please clarify
Discussion. Line 241-248 I appreciate the complexity of the analysis, although the main findings are not very new, in part they were to be expected
Line 247 “attractor” this word is misleading, please change, the same situation with this word in last line of abstract
Line 252-261 Authors should not at all compare results across disciplines because it makes no sense. I suggest to look for other references or write that in football such an analysis has not yet been carried out and these results have no reference in the literature
Line 263-268 I expect this result to be better / more thoroughly discussed
Line 270-282 it seems reasonable to refer here to various tactical systems, e.g. that in the system in which the examined team played 5 playing positions were not specified
Line 329-345 I would suggest authors to combine loose thoughts into broader paragraphs to eliminate short paragraphs that consist of one sentence. This rule should not normally be used (e.g. see the last two sentences / paragraphs in the discussion)
Conclusion and practical application. Legible and a bit short as presented in the main findings in the manuscript

---

## Round 0.2 · Minor Revisions

Thanks for your efforts in addressing the concerns of the two reviewers. While the second reviewer is happy to accept the manuscript I do not quite feel you’ve addressed all of the concerns initially raised by reviewer one. My comments below relate to the lines in the resubmitted PDF.

Line 147 – 149: while your answer in reply to comments regarding the 85% threshold appears detailed, this level detail still does not appear in the revised manuscript please update on the revised manuscript.

Line 388: I also don’t believe the phrase attractor is relevant here please reword this sentence as attractor has a specific skill acquisition context that doesn’t reflect the sentence.

Tables: as requested by reviewer one, there are still insufficient units of measurement attached to the tables. Please update this for table 1 and table 2.

·

Basic reporting

Ok

Experimental design

Ok

Validity of the findings

Ok

Additional comments

Ok

---

## Round 0.3 · Minor Revisions

Thanks for your improvements of the paper. While the extra detail is much appreciated, there may be some minor typographical issues in some of these sentences. Please see some suggestions below:

Procedure: I suggest you make the following changes to the updated paragraph on the 85% calculations. “….in order to delimit the range of activity performed in the SubMIP zone (25). The MIP was calculated with the raw data to which a 10 Hz filter was applied. For the simple variables, standard calculations were applied using the individual player’s 85% of maximum speed from each game and period. Every time that 85% was exceeded, the counting of the event and its duration was calculated. For the complex variables, the previously published formulas were applied to subsequently apply 85%.” In this revised section, please also specify the: 1) type of filter that was applied; 2) provide some examples of the simple variables; 3) provide some examples of the complex variables; and 4) provide a reference to the previously published formulas for these complex variables.

Conclusion: I suggest this be rewritten as “Therefore, we can conclude that physical demands are not determined solely by a player’s position although their position may influence their playing profile during matches”.

Table 1: the fourth dependent variable is listed as m/min (m). This should be described as something like “Relative distance” or “Time normalised distance”, using whatever abbreviation you think is suitable. The units should then be provided as “m/min”.

---

## Round 0.4 · accepted · Accept

I am happy to inform you that the reviewers and I feel that you have addressed our initial constructive criticisms and that your manuscript is acceptable for publication in PeerJ.